# Supporting newly graduated medical doctors in managing COVID-19: An evaluation of a Massive Open Online Course in a limited-resource setting

**Ardi Findyartini**[1,2]\*, **Nadia Greviana**[1,2], **Chaina Hanum**[1], **Joseph Mikhael Husin**[1], **Nani Cahyani Sudarsono**[3,4], **Desak Gede Budi Krisnamurti**[5,6], **Pratiwi Rahadiani**[5]

**1** Medical Education Center, Indonesia Medical Education and Research Institute (IMERI), Faculty of Medicine Universitas Indonesia, Jakarta, Indonesia, **2** Department of Medical Education, Faculty of Medicine Universitas Indonesia, Jakarta, Indonesia, **3** Sports and Exercise Study, Indonesia Medical Education and Research Institute (IMERI), Faculty of Medicine Universitas Indonesia, Jakarta, Indonesia, **4** Department of Community Medicine, Faculty of Medicine Universitas Indonesia, Jakarta, Indonesia, **5** Center for E-learning, Indonesia Medical Education and Research Institute (IMERI), Faculty of Medicine Universitas Indonesia, Jakarta, Indonesia, **6** Department of Pharmacy, Faculty of Medicine Universitas Indonesia, Jakarta, Indonesia

\* ardi.findyartini@ui.ac.id

**Data Availability Statement:** All relevant data can be found here: https://datadryad.org/stash/share/

## Abstract

### Introduction

Newly graduated medical doctors in their internships are positioned to strengthen the front line in combating COVID-19. We developed a Massive Open Online Course (MOOC) to equip them with adequate knowledge for COVID-19 management. This paper aims to analyze the MOOC and evaluate participant satisfaction and increase in knowledge after completing the course.

### Methods

An observational study was conducted. Quantitative data were obtained from questionnaires and pre- and post-tests. Responses to open-ended questions of the questionnaires were collected. Analysis using the Quality Reference Framework was also completed.

### Results

The MOOC consisted of fundamental knowledge of COVID-19 (Part A) and further enrichment (Part B), and the content was written in the Indonesian language. A total of 3,424 and 2,462 participants completed the course in August and November 2020, respectively. Most participants agreed that the platform was easy to navigate, the design was interesting, and the content was aligned with their needs. Pre- and post-test scores in Part A's subjects increased significantly. Factors contributing to and inhibiting usability and areas for improvement were further highlighted.

mvDL4R0c2_-usGV_FkRxTve8gXp4e-p5ilNgxU0DFUY.

**Funding:** The authors received no specific funding for this work.

**Competing interests:** The authors have declared that no competing interests exist.

## Discussion

The use of a specific quality framework facilitated a comprehensive evaluation of the MOOC's strengths, weaknesses, and areas for future improvements. The participants' satisfaction and pre- and post-test results showed that the current MOOC holds great potential benefit for continuing education for medical interns joining the frontliners during the pandemic. Future implementation should consider increasing the quality of learning resources, scaling up the platform and its technical supports, and enhancing organizational supports.

## Introduction

The COVID-19 pandemic has taken a toll in morbidity and mortality as well as disruptive changes and adaptations in everyday life across the world. Public health responses have been conducted globally with different outcomes. Despite a global plan for mass vaccination, the readiness and resilience of health personnel for tackling the overflow of COVID-19 patients in the healthcare setting has never been more critical. Newly graduated medical doctors can strengthen the health personnel team combating COVID-19 at the front line [1,2]. Given that their training does not necessarily provide adequate knowledge on the COVID-19 pandemic and comprehensive management [2], efforts to enhance their knowledge are necessary.

Massive Open Online Courses (MOOCs) have been increasingly used as a convenient platform during the COVID-19 pandemic, given the push to move learning activities online [3]. MOOCs are an online learning tool and have been widely used for distance learning. For the past 10 years, MOOC providers, such as edX, Coursera, and Udacity, have been offering various online courses that can be accessed by millions of learners without geographical boundaries [4]. MOOCs have roots in the value that knowledge should be shared freely without time, demographic, economic, or geographical constraints. Some draw on connectivism theory in their emphasis on discussion forums and collaboration (cMOOC), and others draw on behaviorism theory with emphasis on knowledge sharing and teacher-centric lectures (xMOOC) [5].

Most users in Asian countries use MOOCs as a means to support them to gain specific job skills, prepare for future work, or as part of a professional certification [5]. In the context of medical and health professions education, such an approach holds great potential for facilitating continuing education and professional development [6]. Due to the rapid updates in COVID-19-related knowledge for health personnel and the need to provide flexible access to this knowledge, we developed an online course for newly graduated medical doctors in Indonesia using an xMOOC approach. While open-access resources for the COVID-19 pandemic have been developed worldwide (e.g., WHO, CDC, etc.) [7], the MOOC we developed for new medical graduates adapts the content and delivery modes to the needs of the Indonesian setting. We created *Modul Tanggap Pandemi COVID-19 untuk Dokter Internship Indonesia* (MTPC-I) as an open online course specifically designed for newly graduated medical doctors during the pandemic of COVID-19. This course was designed to equip internship doctors to participate in patient care and treatments for COVID-19 and provide knowledge to increase patient and healthcare workers' safety during the pandemic. This paper aims to analyze MTPC-I using the Quality Reference Framework (QRF) [8] and evaluate MTPC-I participants' satisfaction and increase in knowledge after completion of the course using the Kirkpatrick framework [9]. Since the current MOOC was developed and implemented in a setting with

resource limitations (e.g., unstable internet connections) and widely dispersed users, we also provide evidence on the feasibility of the approach for Continuing Medical Education (CME) in such conditions.

## Methods

The study was approved by the Research Ethics Committee of the Faculty of Medicine Universitas Indonesia, Cipto Mangunkusumo National Referral Hospital (KET-1395/UN2.F1/ETIK/PPM.00.02/2020). The participants provided written consent through the MOOC system which described information that all data obtained from the system will be further analysed by still protecting anonymity and data confidentiality.

### Context

Indonesia has a population of approximately 270 million and is one of the largest archipelagos in the world. Newly graduated medical doctors should complete a 1-year internship program in district hospitals and primary health care centers all over Indonesia before commencing private practice as general practitioners or continuing to specialist programs. The program is organized by the Indonesian Ministry of Health (MoH), and it mandates the interns to complete professional clinical activities with minimum supervision from attending senior medical doctors.

### Design and data collection

An observational study was conducted using quantitative and qualitative data to evaluate the MTPC-I. Quality analysis of the MTPC-I was conducted using the QRF, which consists of the following phases: analysis, design, implementation, realization, and evaluation [8]. Secondary data obtained from the Moodle-based online course platform were used to evaluate users' satisfaction and increase in knowledge. Quantitative data were obtained from the pre- and post-tests and from the user's satisfaction questionnaire, which consisted of seven questions related to the platform and the design used in the module, the learning material and its relevance to the needs in practice. The questionnaire items were validated by the team, were pretested to medical students who completed another MOOC on Covid-19, and revised accordingly. The MOOC platform also allowed us to collect data regarding the time in which participants completed the pre- and post-tests as well as the number of attempts for each test. The data were then used to analyze the access duration of each section. In addition, the questionnaire collected qualitative data in the form of responses to open-ended questions.

### Data analysis

The quality of the MTPC-I was analyzed with a qualitative descriptive approach using the QRF and checklist. All quantitative data obtained from the questionnaires and the pre- and post-test data were analyzed using IBM SPSS Statistic version 21. Descriptive analysis of quantitative and qualitative data was completed. Participants' perceptions of the module implementation in the two batches were compared using Chi-square, Fisher exact analysis. Pre- and post-test data on similar topics in both batches were analyzed with t-dependent or Wilcoxon tests to compare the results across groups. The duration of participants' access to each section was compared with the expected minimum duration estimated by course developers. The total number of attempts on the post-tests was also analyzed descriptively to compare the numbers across sections in Part A. Thematic analysis was conducted to analyze qualitative data obtained from the open-ended questionnaires.

## Results

### Analysis and design phase

MTPC-I was designed through a collaboration between the Medical Education Center (MedEC), the Indonesian Medical Education and Research Institute (IMERI), and the Faculty of Medicine Universitas Indonesia (FMUI). Experts and practitioners across departments in FMUI, including medical specialists of multiple disciplines and public health experts, served as content contributors, facilitators, medical educators, and MOOC providers and designers. MTPC-I was embedded in the Open Course IMERI (OCI), a Moodle-based MOOC platform established by the Center of e-Learning IMERI FMUI (CEL), the providers of this MOOC.

At the beginning of the project, we conducted discussions among all stakeholders to align objectives with those of the MoH. The learners participating in MTPC-I were the newly graduated medical doctors who started medical school in 2013–2014 and finished their clinical training at the beginning of the pandemic. The content of MTPC-I was developed by pulmonologists, internists, neurologists, pediatricians, epidemiologists, microbiologists, ophthalmologists, surgeons, and obstetricians based on the most updated knowledge regarding COVID-19. Resources and materials were aligned with learning outcomes as well as the characteristics of the internship doctors and the educational environment during the internship program. Contributors and facilitators of MTPC-I created new narrative text materials in the form of short literature reviews in the Indonesian language since much of the current literature on COVID-19 was published in English. The course also provided the standard operating procedures of COVID-19, published by the Indonesian medical communities of practices, as primary references to address language barriers.

All interns in the August and November periods of the national internship program were enrolled in MTPC-I. Due to the highly demanding nature of workplace learning during the internship program, we classified the content of MTPC-I into two parts, which were designed sequentially (Fig 1):

1. Part A consisted of fundamental knowledge of COVID-19, curated according to learners' needs and characteristics, and its completion was mandatory.

2. Part B consisted of supplemental and enriching knowledge about COVID-19, and its completion was recommended.

As the main objective of the module was to provide adequate knowledge and cognition, the didactical concept used in MTPC-I was mostly learner-centered using a lectured-based approach. Contributors provided materials mainly in the form of lecture or simulation videos, texts, PowerPoint presentations, reading materials, and discussion forums. Active references were also embedded in the system, allowing learners to click and download for further study. A problem-based approach was used at the end of Part A in which learners were asked to conduct a case study.

To support the flexibility of utilization, MTPC-I was mostly set as individual asynchronous learning with a self-paced approach. Learners could monitor their own learning from a progress bar in the system. Several webinar series were also conducted to facilitate face-to-face interactions with contributors and facilitators. The webinar topics were selected based on the most frequently discussed topics.

Learning assessment was conducted for each topic using pre- and post-tests. Pre- and post-tests were automatically given after participants had completely accessed each topic, and the minimum passing score was 70%. Participants were given unlimited attempts to reach the passing score, and the correct answers were provided after reaching it.

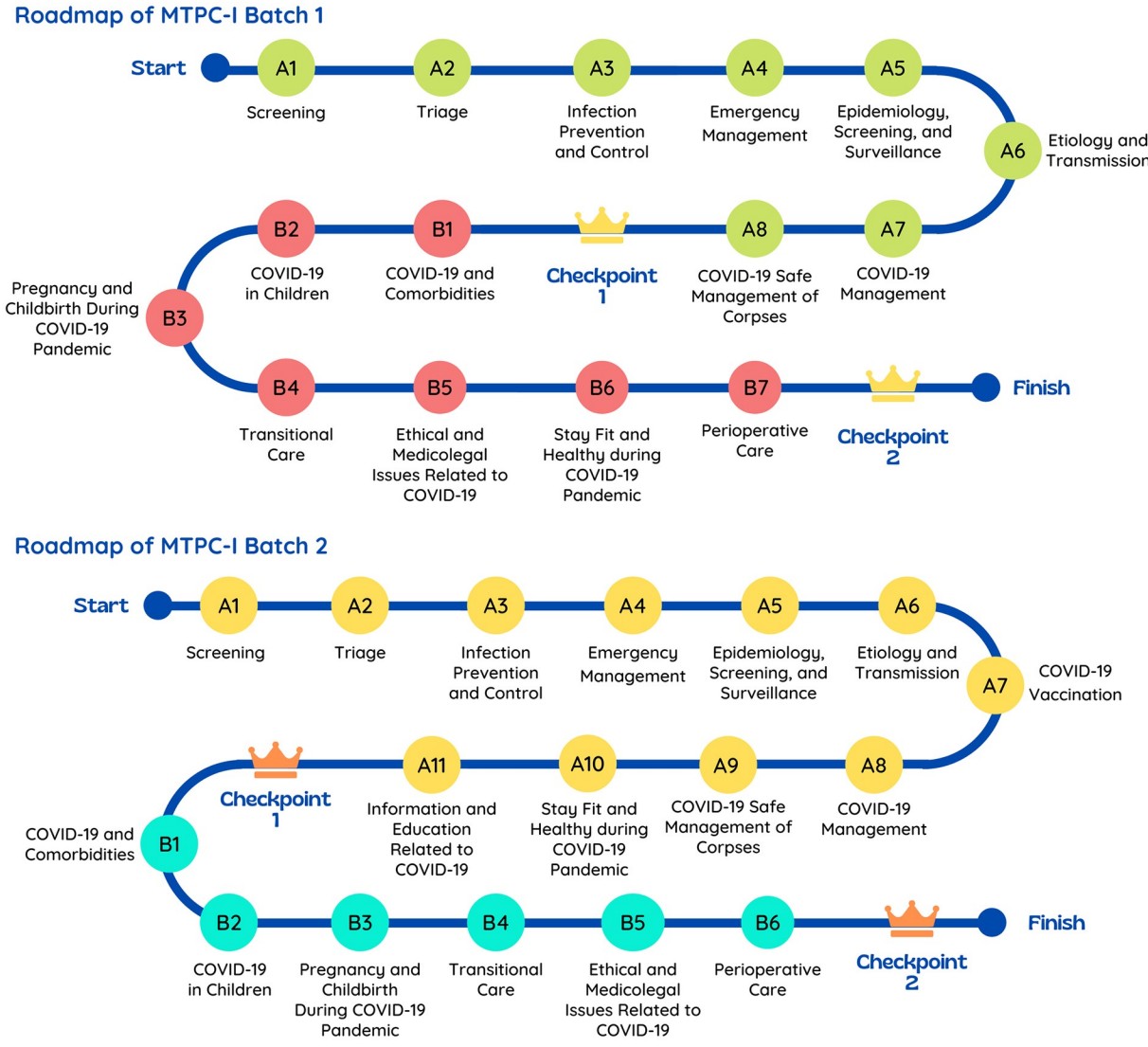

**Fig 1. Content of MTPC-I Batch 1 and Batch 2.**

## Implementation and realization phase

Participants in the Indonesian National Internship Medical Doctors year 2020 were determined by the MoH and automatically enrolled in MTPC-I in two batches:

1. Batch 1, August 2020: 3,424 participants

2. Batch 2, November 2020: 2,462 participants

Given the large number of participants, communication and guidance for implementation were crucial. The MoH announced the mandate to participate in the course, while the MedEC and CEL teams prepared guidelines for logging in to the platform and accessing the module for the first batch. The MedEC team provided a helpdesk officer to assist learners with difficulties logging in to the platform and accessing the course. For second-batch participants, an instructional video was created and uploaded on YouTube for additional technical support.

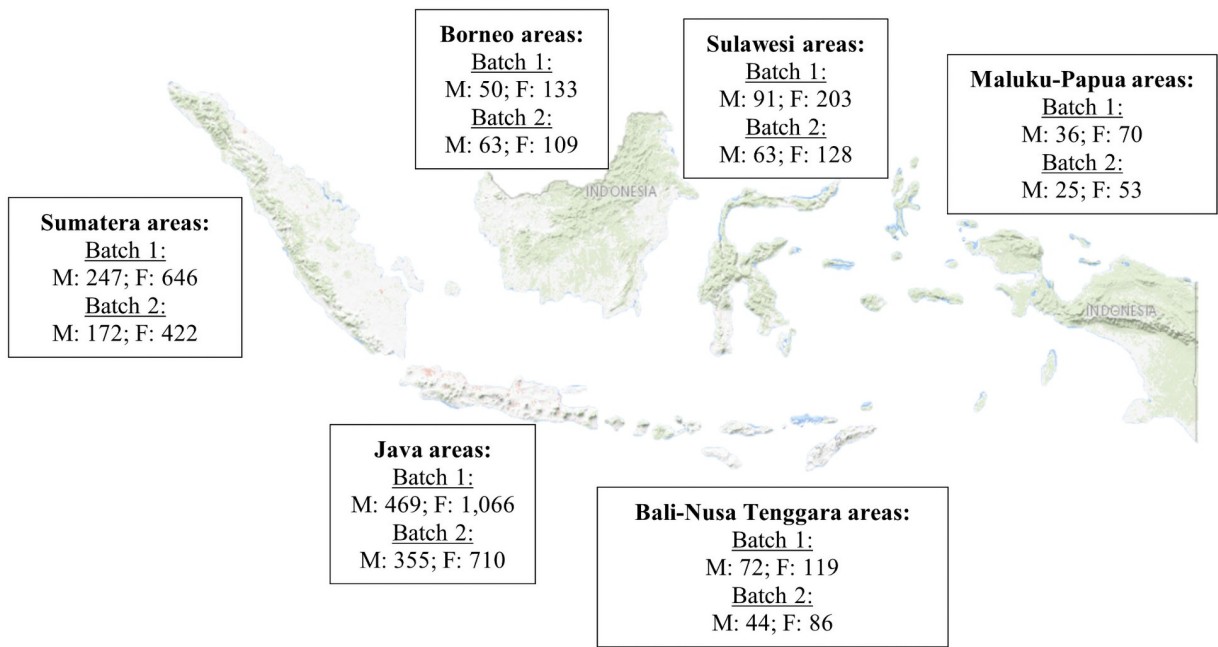

**Fig 2. Characteristics of MTPC-I participants completing Part A from Batch 1 and 2 based on the internship area and gender (source of Indonesian map: https://apps.nationalmap.gov/viewer/).**

The large number of participants simultaneously enrolled in the module also raised some challenges. As designers and providers, the MedEC and CEL teams were aware of the possibility of server overload. To overcome this challenge, the MTPC-I was implemented in stages based on participants' departure schedules. We also prepared a special section at the beginning of the module to give participants the opportunity to try out the main features of the MOOC (e.g., lessons, pages, assignment submissions). Despite these efforts, participants reported that navigation and progress monitoring were challenging as they lacked familiarity with the platform.

We monitored the server during the implementation of MTCP-I, and it loaded a memory usage of 12.8%, and CPU usage was less than 1%. This means that our server was able to handle various types of learning formats and scale up to accommodate many users. A total of 3,202 (93.52%) of Batch 1 participants and 2,230 (90.55%) of Batch 2 participants completed Part A. The geographical distribution and gender of participants who completed Part A are displayed in Fig 2.

Beyond the asynchronous sessions, we realized that participants needed interactions with experts. Therefore, synchronous webinars were conducted to provide this using the Zoom platform and streamed online on YouTube to accommodate the number of participants. Asynchronous access was available if the webinar schedule conflicted with participants' service schedules in the hospitals.

Considering rapid developments in diagnosis, therapy, and healthcare policies regarding COVID-19 as well as participants' needs, course designers and contributors updated the materials continuously during the implementation process. Newly updated topics were highlighted at the end of the courses and were broadcast to all participants with automated messaging in the Moodle-based platform.

## Evaluation phase

We analyzed access duration in each section to evaluate users' participation in the course. We also administered questionnaires on how participants perceived the course at the end of Part A. Apart from the continuous content updates, course designers also evaluated the module questionnaire during the implementation phase and conducted a more thorough evaluation at the end of each course batch. The results of the evaluation phase are presented below.

**Users' participation.**   We analyzed the duration of participants' access in each section and compared it with the expected minimum duration estimated by course developers according to the types of learning activities in each section. The results are shown in Table 1. Overall, the duration of access was lower than the expected minimum duration estimated by course developers. Furthermore, despite the addition of sections in Batch 2, the average access duration of Batch 2 participants was shorter than Batch 1 participants.

**Pre- and post-test scores.**   We analyzed the pre- and post-tests of all sections Part A for each batch to assess the change in the scores. Post-test scores increased significantly for all sections in Batch 1 and 2 ($p < 0.005$), as depicted in Table 2.

**Users' satisfaction questionnaire.**   The results showed that the module provided knowledge and learning experiences related to COVID-19 pandemic that can be of assistance for the internship doctors as frontliners. The participants in MTPC-I Batch 1 and Batch 2 held significantly different perceptions regarding items 4 and 6, as shown in Table 3.

Thematic analysis of the commentaries from the open-ended questions resulted in three main themes: (1) factors contributing to usability, (2) factors inhibiting usability, and (3) areas for improvement.

**Table 1. Access duration in the sections of Part A.**

| No. | Sections | Learning Activities | Expected Minimum Access Duration | Average Access Duration | |
|---|---|---|---|---|---|
| | | | | Batch 1 | Batch 2 |
| 1. | Screening | Video, text text/ embedded guidelines/ slides, references | 3 hours | 1 hours 26 minutes | 1 hours 8 minutes |
| 2. | Triage | Video, text/ embedded guidelines/ slides, references | 3 hours | 1 hour | 48 minutes |
| 3. | Infection prevention and control | Video, text/ embedded guidelines/ slides, references | 3 hours | 1 hour | 42 minutes |
| 4. | Emergency management | Video, text/ embedded guidelines/ slides, references | 2 hours | 55 minutes | 42 minutes |
| 5. | Epidemiology, screening, and surveillance | Text/ embedded guidelines/ slides, references | 2 hours | 30 minutes | 20 minutes |
| 6. | Etiology and transmission | Video, text/ embedded guidelines/ slides, references | 2 hours | 14 minutes | 19 minutes |
| 7. | COVID-19 management | Video, text/ embedded guidelines/ slides, references | 2 hours | 31 minutes | 22 minutes |
| 8. | COVID-19 safe management of corpses | Video, text/ embedded guidelines/ slides, references | 1 hour | 16 minutes | 13 minutes |
| 9. | COVID-19 vaccination | Video, text/ embedded guidelines/ slides, references | 2 hours | | 24 minutes |
| 10. | Staying fit and healthy during the COVID-19 pandemic | Video, text/ embedded guidelines/ slides, references | 2 hours | | 24 minutes |
| 11. | Information and education related to COVID-19 | Video, text/ embedded guidelines/ slides, podcast, references | 2 hours | | 23 minutes |
| **Total Expected Minimum Access Duration** | | | Batch 1: 18 hours Batch 2: 24 hours | 5 hours 52 minutes | 5 hours 45 minutes |

Note: Sections 9, 10, and 11 were unavailable in MTPC-I Batch 1.

**Table 2. Pre- and post-test scores of Part A MTPC-I.**

| No. | Sections | Batch 1 | | | | | | | Batch 2 | | | | | | |
|---|---|---|---|---|---|---|---|---|---|---|---|---|---|---|---|
| | | Pretest | Post-test | Sig. | Attempt (%) | | | | Pretest | Post-test | Sig. | Attempt (%) | | | |
| | | (median (min-max)) | | | 1 | 2 | 3 | >3 | (median (min-max)) | | | 1 | 2 | 3 | >3 |
| 1. | Screening | 8.00 (2.00–10.00) | 10.00 (3.00–10.00) | p = 0.000* | 85.29 | 11.12 | 1.85 | 1.74 | 8.00 (1.00–10.00) | 10.00 (2.00–10.00) | p = 0.000* | 87.07 | 8.81 | 2.14 | 1.98 |
| 2. | Triage | 9.00 (0.00–10.00) | 10.00 (0.00–10.00) | p = 0.000* | 90.31 | 7.51 | 1.21 | 0.97 | 9.00 (0.00–10.00) | 10.00 (2.00–10.00) | p = 0.000* | 91.43 | 5.97 | 0.99 | 1.61 |
| 3. | Infection prevention and control | 8.00 (0.00–10.00) | 10.00 (2.00–10.00) | p = 0.000* | 89.42 | 7.31 | 1.77 | 1.50 | 8.00 (0.00–10.00) | 10.00 (2.00–10.00) | p = 0.000* | 91.97 | 5.23 | 0.91 | 1.90 |
| 4. | Emergency management | 8.00 (0.00–10.00) | 10.00 (2.00–10.00) | p = 0.000* | 89.98 | 7.25 | 1.56 | 1.21 | 8.00 (1.00–10.00) | 10.00 (0.00–10.00) | p = 0.000* | 92.79 | 5.07 | 1.07 | 1.07 |
| 5. | Epidemiology, screening, and surveillance | 8.57 (0.00–10.00) | 10.00 (1.79–10.00) | p = 0.000* | 93.99 | 4.30 | 0.97 | 0.74 | 8.00 (0.00–10.00) | 10.00 (3.00–10.00) | p = 0.000* | 93.94 | 4.66 | 1.07 | 0.33 |
| 6. | Etiology and transmission | 10.00 (0.00–10.00) | 10.00 (6.00–10.00) | p = 0.000* | 98.59 | 1.36 | 0.03 | 0.03 | 10.00 (0.00–10.00) | 10.00 (7.00–10.00) | p = 0.000* | 99.01 | 0.82 | 0.12 | 0.04 |
| 7. | COVID-19 management | 9.00 (2.00–10.00) | 10.00 (0.00–10.00) | p = 0.000* | 95.31 | 4.07 | 0.44 | 0.18 | 8.00 (0.00–10.00) | 10.00 (3.00–10.00) | p = 0.000* | 96.37 | 2.81 | 0.54 | 0.29 |
| 8. | COVID-19 safe management of corpses | 8.00 (0.00–10.00) | 10.00 (0.00–10.00) | p = 0.000* | 87.18 | 7.02 | 2.77 | 3.04 | 9.00 (0.00–10.00) | 10.00 (3.00–10.00) | p = 0.000* | 88.90 | 5.45 | 1.86 | 3.80 |
| 9. | COVID-19 vaccination | | | | | | | | 10.00 (0.00–10.00) | 10.00 (0.00–10.00) | p = 0.000* | 93.65 | 4.70 | 0.91 | 0.74 |
| 10. | Staying fit and healthy during the COVID-19 pandemic | | | | | | | | 9.00 (0.00–10.00) | 10.00 (0.00–10.00) | p = 0.000* | 95.46 | 3.43 | 0.78 | 0.33 |
| 11. | Information and education related to COVID-19 | | | | | | | | 7.00 (0.00–10.00) | 9.00 (1.00–10.00) | p = 0.000* | 88.43 | 6.74 | 1.94 | 2.89 |

Note: Sections 9. 10. and 11 were unavailable in MTPC-I Batch 1.

1. Factors contributing to usability

The participants appreciated the online learning course provided in MTPC-I because it could be accessed flexibly from different locations and did not require direct face-to-face interaction. The module interface was interesting and user-friendly. Additionally, participants emphasized that the material selection, such as text, narratives, infographics, and videos, allowed them to choose the materials according to their preference, which facilitated their learning.

*This online course is one of the best options [I have], considering that the learning process does not require face-to-face interaction. The materials covered in this module are essential for doctors working in primary health care during this pandemic.*

(F, Java, Batch 2)

**Table 3. Users' satisfaction questionnaire on the MTPC-I.**

| No. | Questions | Batch 1 (%) | | | | Batch 2 (%) | | | | Chi-square | |
|---|---|---|---|---|---|---|---|---|---|---|---|
| | | Disagree | | Agree | | Disagree | | Agree | | $X^2$ (df) | Sig. |
| | | SD | D | A | SA | SD | D | A | SA | | |
| 1. | The OCI platform was easy to use. | 2.6 | 3.0 | 57.2 | 37.2 | 2.6 | 2.7 | 55.1 | 39.6 | .126 (1) | p = 0.722 |
| 2. | The arrangement of the content in the module was easy to learn. | 2.6 | 2.7 | 57.3 | 37.4 | 2.2 | 2.0 | 57.0 | 38.7 | 2.945 (1) | p = 0.086 |
| 3. | The learning outcome and the content of the module were in accordance with the need in society. | 2.5 | 0.7 | 57.0 | 39.8 | 2.2 | 0.5 | 54.4 | 42.8 | 1.040 (1) | p = 0.308 |
| 4. | The pre- and post-tests correspond to the learning materials used in each topic. | 2.6 | 5.4 | 57.6 | 34.4 | 2.2 | 2.0 | 55.7 | 40.1 | 32.022 (1) | p = 0.000* |
| 5. | Duration of completion of the module was equal to the estimated duration described in the module. | 2.6 | 3.7 | 63.1 | 30.6 | 2.2 | 3.2 | 60.5 | 34.1 | 1.774 (1) | p = 0.183 |
| 6. | The design of the module got me interested in learning more. | 2.7 | 4.8 | 58.2 | 34.4 | 2.4 | 3.7 | 56.8 | 37.2 | 3.909 (1) | p = 0.048* |
| 7. | The module gave me enough learning experiences to be used as a COVID-19 frontliner. | 2.5 | 1.0 | 55.0 | 41.6 | 2.3 | 0.5 | 53.1 | 44.0 | 1.588 (1) | p = 0.208 |

* statistically significant, p < 0.05.

**SA: Strongly Agree, A: Agree, D: Disagree, SD: Strongly Disagree.

*I think this online module is easy to follow since [all sections] are completed with adequate and clear explanations, audio-visual materials, and necessary figures.*

(M, Sumatera, Batch 1)

2. Factors inhibiting usability

Despite these contributing factors, some factors that inhibit usability were also suggested by the participants—for example, the need to increase audio-visual material quality and facilitate easier navigation of the platform. The participants were also concerned about the required time to complete each section, which they hoped could be set more flexibly given their internship work assignment. Unstable internet connection in some internship locations was also identified as a challenge that could inhibit the usability of the MTPC-I, particularly for accessing video content, which required a large bandwidth.

*At the beginning, I was not able to access and utilize the resources in the module because I was not familiar with the platform. It would be better if there is a video tutorial explaining the details on how to navigate the module platform.*

(F, Sumatera, Batch 1)

*Most of us had trouble accessing the module from remote sites where the internet connection is unstable.*

(F, Sumatera, Batch 2)

3. Areas for improvement

Participants also suggested further areas for improvement. For example, subjects such as vaccination, public education on COVID-19, and how to maintain one's own health during the pandemic were suggested by Batch 1 participants and added to the content for MTPC-I Batch 2. Further input on the subjects included management of healthcare facilities during the pandemic in a limited-resource setting, which will be taken into account for future development in MTPC-I. Participants also made suggestions for enhancing the quality of learning resource materials, including animated videos, shorter videos, better aligned pre- and post-test questions, and more clinical cases for exercise.

*Some videos are too long and a bit boring. [I think] the video should be arranged in chunks, be shorter, and be completed with more interesting animations.*

(F, Java, Batch 1)

*More case applications in the pre- and post-tests will be good. Better aligning the pre- and post-tests with the learning resources is also necessary.*

(F, Java, Batch 1)

"*The module provides thorough knowledge on COVID-19. It would be better if it also gives us [the interns] information on how to manage COVID-19 in rural areas or in a limited resource setting.*"

(F, Sumatera, Batch 2)

## Discussion

This paper aimed to analyze MTPC-I using the Quality Reference Framework (QRF) [8] and evaluate MTPC-I participants' satisfaction and increase in knowledge following the completion of the course using the Kirkpatrick framework [9]. Since MTPC-I with xMOOC format [5] was part of the in-service training and CME for newly graduated medical doctors, we found that the MOOC was very timely and strategic to equip the graduates who were geographically disperse with necessary knowledge for working on the front line during the pandemic.

There have been concerns about misinformation and disinformation regarding COVID-19 for both health professionals and the general population due to the rapid growth of evidence in all aspects of COVID-19, including epidemiology, virology, pathogenesis, diagnosis and treatment, public education, and communication [7]. Therefore, during the analysis stage, we deliberately considered five critical needs: a. minimum level of knowledge for the interns as frontline team members, b. responsive, easily accessible, online course suitable for a large number of participants who were geographically distributed, c. the nature of the COVID-19 pandemic-related subject content, d. module completion arrangement that allows for progressive and flexible completion for the interns juggling with their new work environment, and e. the new graduates as a modern generation with particular learning needs and preferences. Following the QRF criteria for the first two stages, we analyzed our stakeholders, targeted learners, learning outcomes, content and materials, media, technical aspect, interaction and feedback concept, and assessment component [8] and incorporated them in the MTPC-I design. While we realized the need to keep improving in all areas, two main aspects that we found would require more detailed preparation and design were related to the organizational role, resources, and budget planning since two institutions were involved (MoH and FMUI). Given our aim to provide a fast response, we took immediate decisions to utilize available resources, including resource persons from different departments in the medical school. For this MOOC to be sustained, the latter aspects should also be carefully considered in the future.

Given the different preferred learning styles of MOOC participants, it was important for designers and providers to build pedagogical hypotheses on how participants learn in the MOOC [10]. Therefore, MTPC-I was developed using various didactical approaches suitable for our target learners, including learner-centered, competence-based, task-based, and interactive approaches with variations of video lecturing, narrative texts, embedded PowerPoint presentations, and downloadable references. In this way, we were able to accommodate different learning styles among participants and increase participation. The problem-based approach

was also used in formative assessments using pre- and post-tests and case studies. However, learning activities promoting participation in the learning environment and interaction between participants were identified as a challenge in MTPC-I implementation, where they were limited to the discussion forum and live webinar series [8,11]. The limited use of learning activities that induced participation and interaction may have also played a role in the short average access duration among participants in both batches. The decreased average access duration of the course in Batch 2, despite the increase in learning sections, was also caused by the fact that the second batch took the online course after the pandemic had been going on for almost one year. Hence, some basic knowledge provided in the online course was already known from other open resources. This demonstrates the need for course providers to reprioritize and rapidly adjust the content regularly. Furthermore, the development of MTPC-I should include the use of social learning interventions, such as small group projects in which participants apply the course content to their context [12], which were possible as national internship programs were conducted in groups.

Furthermore, considering the development, implementation, and evaluation stages [8], it can be highlighted that MTPC-I was used to address the immediate need of the continuing medical education program for newly graduated medical doctors in Indonesia [6]. With its ability to reach large numbers of participants, the MOOC course was attractive to Indonesian policy makers to equip the newly graduated medical doctors with sufficient knowledge about the COVID-19 as it emerged worldwide. With a completion rate of the mandatory part (Part A) of approximately 90–93% and a significant increase in participants' knowledge, this study showed that MTPC-I was successfully used in an archipelagic country with various cultures, geographic patterns, and diverse resources.

The high level of completion was related to external motivation factors. Part A was mandatory and certificates of completion were provided for participants who successfully completed it; such recognition increases participation and completion levels in MOOCs [13]. The MoH required participants to upload their certificates to the system as a prerequisite for completing the national internship program. Culture is also known to influence participation in online learning in areas including language proficiency, familiarity with computers and systems, and learning habits (e.g., teacher-centered, student-centered) [14]. Considering the variation in English proficiency among participants, the use of Indonesian language, the native language, as the main language of instruction both in the Learning Management System (LMS) and the content of MTPC-I also played a role in increasing participation and the level of completion [15]. The use of Bahasa Indonesia in the MTPC-I was strategic since learners in MOOCs in their native language report a greater growth in knowledge; thus, the significant increase in participants' knowledge found in this study might also be related to the support of the language of the learning resources [16].

The qualitative reports in the questionnaire identified some factors as inhibiting use of the MOOC, including the limited internet access in some regions, especially in remote areas. Limited internet access as a drawback of the use of MOOCs for CME has been similarly reported in other developing countries [15,17,18]. The use of varied forms of content, not only video lectures that use large internet bandwidth, was helpful.

Beyond the support for language literacy and pedagogical aspects, MTPC-I designers also provided technical support to address the lack of technical skills and familiarity with the LMS among participants. The strength of MTPC-I lies in the co-construction of knowledge through curation and updating of the rapid information influx about COVID-19 as well as offering a variety of learning resources and strong collaboration among MOOC facilitators, designers, and providers to support participants' learning. We are aware that the real-time monitoring of server performance that we have now is not sufficient to support the implementation of

MOOCs in the future, which will continue to experience high access levels. Moodle's design, which allows for strongly scalable setups, will be favorable for improving our MOOC service quality. We also note the importance of optimizing the MTPC-I feature to document the participants' learning processes in more details such as time of viewing in each section and further consider it in issuing certificate of completion in the future.

A technical support team maintained the MOOC system periodically, and they identified, analyzed, and then solved the problems in time so that the system could run efficiently. Technical support response is reported to influence users to adopt MOOC platforms [19,20]. Moreover, Chen et al. (2020) also state that providing comprehensive, timely, and convenient response support has a positive effect on participants' attitudes towards learning [21].

This study highlights several impacts. First, the main impact was seen in the sudden involvement in an unprecedented scale of online learning, which required many experts working collectively to include all necessary knowledge in learning activities for the frontliners in minimum time [22]. Experts' involvement from across medical departments in preparing newly graduated medical doctors using a MOOC should also be considered a fulfillment of their academic roles. Second, as the lives of newly graduated medical doctors were disrupted during this pandemic, the pedagogical approach of online education used in this study helped them to adapt with resilience and participate in a CME program [23,24]. Therefore, the impact of this approach should be considered in the context of the pandemic situation. Third, this study showed that the development and implementation of a MOOC in a limited-resource setting was feasible, even though it might not be well comprehended. This underscores the need for better planning for future development and the importance of technical support for learners with different levels of familiarity with MOOCs. The impact of the pandemic has changed how faculty members should be working collectively within and across institutions to develop a meaningful online learning environment to support medical practitioners' lifelong learning.

This study has several limitations. Regardless of the need to serve the whole country, this study is based on one institution's experience, and it may be challenging to infer the results to other institutions. The fast response required was not an ideal situation to test for inter-institution collaboration. Furthermore, the lack of data reflecting more details of the interns' situations constrained our ability to demonstrate the strength of MOOC support in times of need. Finally, despite our aim to utilize the QRF framework to evaluate the MTPC-I comprehensively in terms of the program processes, we realize that current data indicate outcomes in satisfaction and learning levels [9]. Further evaluations of the knowledge transfer to implementation in practice were not yet covered.

## Conclusion

This study has elaborated the design, implementation, and evaluation of MTPC-I as a MOOC platform for new medical graduates working as interns in geographically disperse areas during the COVID-19 pandemic. The use of a quality framework specific for MOOCs facilitated a comprehensive evaluation of our MOOC focused on the strengths, weaknesses, and areas for future improvement. The participants' satisfaction and pre- and post-test results showed that the MTPC-I was an effective means for delivering CME to the interns joining the front lines during the pandemic. Its impact was supported by the fast response of multidisciplinary experts and team members, the availability and accessibility of the MOOC platform, and the use of the Indonesian language in the learning resources. Future implementation of the MOOC should consider increasing the quality of learning resources, scaling up the platform and its technical supports, and the organizational and administrative support required from the institution.

## Acknowledgments

The authors would like to acknowledge the Ministry of Health, Republic of Indonesia for the trust that has been given to develop and organize MTPC-I for the national internship program participants. We also would like to thank all Indonesian national internship program participants for their participation in this study. We also expressed high gratitude for the Board of Directors of Indonesian Medical Education and Research Institutes (IMERI) Faculty of Medicine Universitas Indonesia and Dean and Vice Deans of Faculty of Medicine Universitas Indonesia for the continuous support. We also would like to highly appreciate the extended support from Prof Ratna Sitompul, Prof Akmal Taher, Prof Pratiwi Sudharmono and Dr Ponco Birowo of Yayasan Sri Oemijati during the development and implementation of MTPC-I.

## Author Contributions

**Conceptualization:** Ardi Findyartini, Nadia Greviana, Nani Cahyani Sudarsono.

**Data curation:** Ardi Findyartini, Nadia Greviana, Joseph Mikhael Husin.

**Formal analysis:** Ardi Findyartini, Nadia Greviana, Chaina Hanum, Joseph Mikhael Husin, Nani Cahyani Sudarsono, Desak Gede Budi Krisnamurti.

**Investigation:** Ardi Findyartini, Nadia Greviana, Chaina Hanum, Joseph Mikhael Husin.

**Methodology:** Ardi Findyartini, Nadia Greviana, Chaina Hanum, Joseph Mikhael Husin.

**Project administration:** Ardi Findyartini, Nadia Greviana, Pratiwi Rahadiani.

**Resources:** Ardi Findyartini, Nadia Greviana, Nani Cahyani Sudarsono, Desak Gede Budi Krisnamurti, Pratiwi Rahadiani.

**Software:** Desak Gede Budi Krisnamurti, Pratiwi Rahadiani.

**Supervision:** Ardi Findyartini, Nani Cahyani Sudarsono.

**Visualization:** Chaina Hanum.

**Writing – original draft:** Ardi Findyartini, Nadia Greviana, Chaina Hanum, Joseph Mikhael Husin, Nani Cahyani Sudarsono, Desak Gede Budi Krisnamurti, Pratiwi Rahadiani.

**Writing – review & editing:** Ardi Findyartini, Nadia Greviana, Nani Cahyani Sudarsono.

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
