## [Decision Letter · Decision Letter 0]

22 May 2021

PONE-D-21-12216

Supporting newly graduated medical doctors in managing COVID-19: An evaluation of Massive Open Online Course in a limited resource setting

PLOS ONE

Dear Dr. Findyartini,

Thank you for submitting your manuscript to PLOS ONE. After careful consideration, we feel that it has merit but does not fully meet PLOS ONE’s publication criteria as it currently stands. Therefore, we invite you to submit a revised version of the manuscript that addresses the points raised during the review process.

We look forward to receiving your revised manuscript.

Kind regards,

M. Usman Ashraf, Ph.D

Academic Editor

PLOS ONE

Journal Requirements:

Please include additional information regarding the survey or questionnaire used in the study and ensure that you have provided sufficient details that others could replicate the analyses. For instance, if you developed a questionnaire as part of this study and it is not under a copyright more restrictive than CC-BY, please include a copy, in both the original language and English, as Supporting Information. Moreover, please include more details on how the questionnaire was pre-tested, and whether it was validated.

4. We note that Figure 2 your submission contain map images which may be copyrighted. All PLOS content is published under the Creative Commons Attribution License (CC BY 4.0), which means that the manuscript, images, and Supporting Information files will be freely available online, and any third party is permitted to access, download, copy, distribute, and use these materials in any way, even commercially, with proper attribution. For these reasons, we cannot publish previously copyrighted maps or satellite images created using proprietary data, such as Google software (Google Maps, Street View, and Earth). For more information, see our copyright guidelines: http://journals.plos.org/plosone/s/licenses-and-copyright.

1.              You may seek permission from the original copyright holder of Figure 2 to publish the content specifically under the CC BY 4.0 license. 

5. Please upload a copy of Supporting Information Figures 1-2 and Table which you refer to in your text on page 24.

Reviewers' comments:

Reviewer's Responses to Questions

**Comments to the Author**

1. Is the manuscript technically sound, and do the data support the conclusions?

Reviewer #1: Yes

Reviewer #2: Yes

2. Has the statistical analysis been performed appropriately and rigorously? 

Reviewer #1: Yes

Reviewer #2: Yes

3. Have the authors made all data underlying the findings in their manuscript fully available?

Reviewer #1: Yes

Reviewer #2: Yes

4. Is the manuscript presented in an intelligible fashion and written in standard English?

Reviewer #1: Yes

Reviewer #2: Yes

5. Review Comments to the Author

Reviewer #1: 1. The manuscript is well drafted and highlights the stength of the MOOC for conducting courses on medical science.

2. Evaluation of the audiance attentiveness should also be highlighted i.e. how attentive were the participants during online classes.

3. Time of viewing the online lecture should be incorporated for issuing the certificate of completion.

4. Both the figures provided are not readable. Improve the figure quality.

Reviewer #2: First, I would like to thank the authors for their initiative role to give a hand facing the COVID-19 pandemic. I checked the course website. It is very user-friendly, organized, and well-prepared.

Generally, the manuscript is clear and accepted, but there are a few questions and comments:

1. How the authors justify the difference between the number of participants in the two batches?

2. line 72-73: the sentence shows MOOC as if it just emerged during this pandemic, and it is not.

It should be rephrased.

3. Line 194: authors gave "unlimited opportunity".

Did you know how many opportunities were used for each section to be passed?

I think this could indicate the difficulty of each section and refer to some potential improvements.

4. In the third theme (Line 288):

The authors did not report it related to Batch 2 as well. Didn't you receive any suggestions from this batch?

5. line 321 - 327: Authors showed the critical needs they considered in THEIR work, but there is a reference at the end of the sentences!!

I cross-checked this reference, but I think it is not clear what they are referring to.

6. There are very frequent grammar and punctuation mistakes, for example, line 37, 43, 44, 48, 55, 57, 66,68, 78,..... and almost all over the manuscript.

It is recommended to have proofreading revision by a native English speaker.

7. Few references are in different styles as in line 71,.... 

Also, some hyperlinks in the reference list are not working

Looking forward to reading your revised manuscript

6. PLOS authors have the option to publish the peer review history of their article (what does this mean?). If published, this will include your full peer review and any attached files.

Reviewer #1: **Yes: **Kamal Pandey

Reviewer #2: **Yes: **Hebatullah Abdulazeem

---

## [Author Response · Author response to Decision Letter 0]

6 Jul 2021

Dear Chief Editor and Reviewers of PLOS ONE,

On behalf of the authors, I would like to extend our appreciation and gratitude towards positive and encouraging feedback on our manuscript submission titled above. We have tried to revise the manuscript accordingly, with the details as uploaded in the response to reviewers file. We hope that the revisions are as expected, and the revised manuscript can be published on the PLOS ONE journal.

---

## [Decision Letter · Decision Letter 1]

23 Aug 2021

Supporting newly graduated medical doctors in managing COVID-19: An evaluation of Massive Open Online Course in a limited resource setting

PONE-D-21-12216R1

Dear Dr. Findyartini,

We’re pleased to inform you that your manuscript has been judged scientifically suitable for publication and will be formally accepted for publication once it meets all outstanding technical requirements.

Kind regards,

M. Usman Ashraf, Ph.D

Academic Editor

PLOS ONE

Additional Editor Comments (optional):

Reviewers' comments:

Reviewer's Responses to Questions

**Comments to the Author**

1. If the authors have adequately addressed your comments raised in a previous round of review and you feel that this manuscript is now acceptable for publication, you may indicate that here to bypass the “Comments to the Author” section, enter your conflict of interest statement in the “Confidential to Editor” section, and submit your "Accept" recommendation.

Reviewer #1: All comments have been addressed

Reviewer #2: All comments have been addressed

2. Is the manuscript technically sound, and do the data support the conclusions?

Reviewer #1: Yes

Reviewer #2: Yes

3. Has the statistical analysis been performed appropriately and rigorously? 

Reviewer #1: Yes

Reviewer #2: Yes

4. Have the authors made all data underlying the findings in their manuscript fully available?

Reviewer #1: Yes

Reviewer #2: Yes

5. Is the manuscript presented in an intelligible fashion and written in standard English?

Reviewer #1: Yes

Reviewer #2: Yes

6. Review Comments to the Author

Reviewer #1: All the comments offered in review one have been well addressed, the comment on certification based on the viewing time have been addressed by incorporating as future scope in the discussion section. Qualitu of the figures have also been improved. The data used for the analysis is now available as a public link

Reviewer #2: This paper addressed an Indonesia-produced massive open online course (MOOC), MTPC-1, that supported educating the newly graduated medical doctors about COVID-19. Using the native language of the participants, the Indonesian language, provided a greater growth in knowledge. This course was analyzed using the Quality Reference Framework (QRF), which consists of analysis, design, implementation, realization, and evaluation steps. Pre- and Post-test design showed a statistical significant improvement of the participants' knowledge.

Further production of MOOCs of higher quality of learning sources, with matched technical support, and targeting the specific needs of course's participants, such as language, level of knowledge, and expected outcome are recommended.

However, the authors did not use a reporting guideline for observational studies (this is recommended for the future collaborations), it is still a well-conducted study with clear objectives, statistical analysis, and interpretation.

7. PLOS authors have the option to publish the peer review history of their article (what does this mean?). If published, this will include your full peer review and any attached files.

Reviewer #1: **Yes: **Kamal Pandey

Reviewer #2: **Yes: **Hebatullah Abdulazeem

---

## [Editor Report · Acceptance letter]

3 Sep 2021

PONE-D-21-12216R1 

Supporting newly graduated medical doctors in managing COVID-19: An evaluation of a Massive Open Online Course in a limited-resource setting 

Dear Dr. Findyartini:

I'm pleased to inform you that your manuscript has been deemed suitable for publication in PLOS ONE. Congratulations! Your manuscript is now with our production department. 

Kind regards, 

on behalf of

Dr. M. Usman Ashraf 

Academic Editor

PLOS ONE